



# Observing atmospheric formaldehyde (HCHO) from space: validation and intercomparison of six retrievals from four satellites (OMI, GOME2A, GOME2B, OMPS) with SEAC[4]RS aircraft observations over the Southeast US

Lei Zhu[1], Daniel J. Jacob[1,2], Patrick S. Kim[2], Jenny A. Fisher[3,4], Karen Yu[1], Katherine R. Travis[1], Loretta J. Mickley[1], Robert M. Yantosca[1], Melissa P. Sulprizio[1], Isabelle De Smedt[5], Gonzalo Gonzalez Abad[6], Kelly Chance[6], Can Li[7,8], Richard Ferrare[9], Alan Fried[10], Johnathan W. Hair[9], Thomsa F. Hanisco[8], Dirk Richter[10], Amy Jo Scarino[11], James Walega[10], Petter Weibring[10], Glenn M. Wolfe[8,12]

[1]John A. Paulson School of Engineering and Applied Sciences, Harvard University, Cambridge, MA, USA
[2]Department of Earth and Planetary Sciences, Harvard University, Cambridge, MA, USA
[3]Centre for Atmospheric Chemistry, School of Chemistry, University of Wollongong, Wollongong, NSW, Australia
[4]School of Earth and Environmental Sciences, University of Wollongong, Wollongong, NSW, Australia
[5]Belgian Institute for Space Aeronomy (BIRA-IASB), Brussels, Belgium
[6]Harvard-Smithsonian Center for Astrophysics, Cambridge, MA, USA
[7]Earth System Science Interdisciplinary Center, University of Maryland, College Park, Maryland, USA
[8]NASA Goddard Space Flight Center, Greenbelt, Maryland, USA
[9]NASA Langley Research Center, Hampton, VA 23681, USA
[10]Institute of Arctic and Alpine Research, Univ. of Colorado, Boulder, CO, USA
[11]Science Systems and Applications, Inc., Hampton, VA, USA
[12]Joint Center for Earth Systems Technology, University of Maryland Baltimore County, Baltimore, Maryland, USA

*Correspondence to*: Lei Zhu (leizhu@fas.harvard.edu)

**Abstract.** Formaldehyde (HCHO) column data from satellites are widely used as a proxy for emissions of volatile organic compounds (VOCs), but validation of the data has been extremely limited. Here we use highly accurate HCHO aircraft
observations from the NASA SEAC[4]RS campaign over the Southeast US in August–September 2013 to validate and intercompare six operational and research retrievals of HCHO columns from four different satellite instruments (OMI, GOME2A, GOME2B and OMPS) and three different research groups. The GEOS-Chem chemical transport model provides a common intercomparison platform. We find that all retrievals capture the HCHO maximum over Arkansas and Louisiana, reflecting high emissions of biogenic isoprene, and are consistent in their spatial variability over the Southeast US ($r$=0.4–0.8
on a $0.5^{\circ} \times 0.5^{\circ}$ grid) as well as their day-to-day variability ($r$=0.5–0.8). However, all satellite retrievals are biased low in the mean by 20–51%, which would lead to corresponding bias in estimates of isoprene emissions from the satellite data. The smallest bias is for OMI-BIRA, which has the highest corrected slant columns and the lowest scattering weights in its air mass factor (AMF) calculation. Correcting the assumed HCHO vertical profiles (shape factors) used in the AMF calculation would further reduce the bias in the OMI-BIRA data. We conclude that current satellite HCHO data provide a reliable proxy





for isoprene emission variability but with a low mean bias due both to the corrected slant columns and the scattering weights used in the retrievals.

## 1 Introduction

Formaldehyde (HCHO) is a high-yield product from the atmospheric oxidation of volatile organic compounds
(VOCs). Methane oxidation largely defines the tropospheric HCHO background. Higher HCHO concentrations over continents are due to short-lived non-methane VOCs (NMVOCs). Loss of HCHO is mainly by photolysis and oxidation by OH, resulting in an atmospheric lifetime on the order of hours. HCHO is detectable from space by solar UV backscatter between 325 and 360 nm [Chance et al, 2000]. HCHO column data from satellites have been used in a number of studies as top-down constraints on NMVOC emissions from biogenic, anthropogenic, and open fire sources [Palmer et al 2003; Shim
et al 2005; Stavrakou et al 2009; Marais et al 2012; Barkley et al., 2013; Zhu et al., 2014]. However, the satellite data have received little validation so far. Here we validate and intercompare six different HCHO retrievals from four satellites instruments (OMI, GOME2A, GOME2B, OMPS) and three different groups with aircraft observations from the NASA SEAC$^4$RS (Studies of Emissions, Atmospheric Composition, Clouds and Climate Coupling by Regional Surveys) campaign over the Southeast US in summer 2013 [Toon et al., 2015].

HCHO columns (molecules cm$^{-2}$) have been continuously observed from space since GOME (1996–2003; Chance et al. [2000]) and SCIAMACHY (2003–2012; Wittrock et al. [2006]). Observations are presently available from OMI (2004–), GOME2A (2006–), OMPS (2011–), and GOME2B (2012–). A slant column density along the atmospheric path of the solar radiation, propagating through the atmosphere and back-scattered to the satellite, is retrieved from the satellite instruments. Conversion to a vertical column is done with an air mass factor (AMF) that depends on the satellite viewing
geometry, the surface albedo, the vertical HCHO profile, and the vertical distributions of clouds and aerosols [Palmer et al., 2001]. Scattering by air molecules causes the AMF to be highly sensitive to the HCHO vertical distribution, which has to be independently specified [Hewson et al., 2015].

Validation of HCHO satellite data sets has been extremely limited due to (1) the large noise in individual satellite retrievals, requiring extensive data averaging to enhance detection; and (2) the limited number of HCHO measurements
acquired from aircraft or from the ground. Martin et al. [2004] validated GOME HCHO columns with aircraft observations in eastern Texas averaged over two campaigns (June–July 1999 and August–September 2000), and found GOME to be too high by 16% on average. Comparison of SCIAMACHY data to ground-based measurements of HCHO columns found no significant mean bias [Wittrock et al., 2006; Vigouroux et al., 2009]. Validation with remotely sensed vertical profiles indicates a 20–40% underestimate in OMI and GOME2 data [De Smedt et al. 2015].
The SEAC$^4$RS campaign offers an exceptional opportunity for validating satellite HCHO data. HCHO columns over the Southeast US in summer are among the highest in the world [Kurosu et al., 2004], due to large emissions of biogenic isoprene from vegetation [Guenther et al., 2006]. Several studies have used HCHO data from space as constraints




on isoprene emission in the Southeast US [Palmer et al., 2006; Millet et al., 2008; Valin et al., 2016]. The SEAC[4]RS aircraft payload included two independent in situ HCHO measurements: the Compact Atmospheric Multispecies Spectrometer (CAMS) [D. Richter et al., 2015] and the NASA GSFC In Situ Airborne Formaldehyde (ISAF) [Cazorla et al., 2015].

5   The SEAC[4]RS aircraft did not conduct direct satellite validation profiles, nor would these be helpful because of the large noise in individual retrievals [A. Richter et al., 2013]. Instead we use here an indirect validation method involving joint comparisons of satellite and in situ HCHO observations with the GEOS-Chem chemical transport model (CTM; Bey et al., [2001]). Satellite and in situ observations do not need to be concurrent, thus increasing considerably the range of data and conditions that can be used for validation.

## 2 Satellite data sets

10   Table 1 lists the six different satellite retrievals of HCHO produced during the SEAC[4]RS campaign. These are from four satellite instruments (OMI, GOME2A, GOME2B, OMPS) on different platforms, with retrievals produced by independent groups for OMI and OMPS. OMI, flown on the NASA Aura research satellite, has much higher spatial resolution than the other instruments. GOME2A and GOME2B are the first successive instruments of a long-term operational commitment by the EUMETSAT European agency for observing atmospheric composition from space [Callies

15 et al., 2000]. OMPS is the first instrument of a similar long-term operational commitment by NOAA in the US [Dittman, et al., 2002].

  All instruments in Table 1 provide dense data sets, with full coverage of the Earth's surface in 1 day for OMI and OMPS, 3 days for GOME2A (since July 2013), and 1.5 days for GOME2B. Except for OMPS-PCA, single-scene detection limits ($0.5$–$1.0\times10^{16}$ molecules cm$^{-2}$) are determined by uncertainties in fitting the backscattered solar spectra. AMFs add

20 another error of 30–100% for single-scene retrievals [Gonzalez Abad et al., 2015a]. Total error on monthly means reduces to 20–40% ($0.1$–$0.3\times10^{16}$ molecules cm$^{-2}$ for the Southeast US) [De Smedt et al., 2008]. OMPS-PCA has a single-scene detection limit of $1.2\times10^{16}$ molecules cm$^{-2}$ determined as the 4-times of 1-sigma noise over the Pacific Ocean [Li et al., 2015]. Here and elsewhere, we use only satellite pixels with solar zenith angle less than 60$^{\circ}$, cloud fraction less than 0.3, and row anomalies (for OMI) screened.

25   All retrievals (except OMPS-PCA) fit the slant column density (SCD) of HCHO from the backscattered solar radiance spectra and then subtract the SCD over the remote Pacific for the same latitude and observing time to remove offsets [Khokhar et al., 2005]. The resulting corrected SCD ($\Delta\Omega_S$) thus represents a HCHO enhancement over the Pacific background. $\Delta\Omega_S$ is converted to the HCHO vertical column density (VCD, $\Omega$) by applying an air mass factor (*AMF*) and a background correction ($\Omega_o$):

30   $\Omega = \dfrac{\Delta\Omega_S}{AMF} + \Omega_o.$          (1)

The background correction $\Omega_o$ is the HCHO column simulated by a CTM (Table 1) for the remote Pacific at the





corresponding latitude and observing time. For OMPS-PCA, it derives the VCD in one step using spectrally varying Jacobians [Li et al., 2015].

The AMF depends on the solar zenith angle ($\theta_Z$) and satellite viewing angle ($\theta_V$), on the scattering properties of the atmosphere and the surface, and on the vertical profile of HCHO concentration. It is computed following Palmer et al. [2001],

as the product of a geometrical AMF ($AMF_G$) describing the viewing geometry in a non-scattering atmosphere, and a correction with scattering weights $w$ applied to the vertical shape factors $S$:

$$AMF_G = \frac{1}{\cos\theta_Z} + \frac{1}{\cos\theta_V}. \tag{2}$$

$$\mathrm{AMF} = AMF_G + \int_{P_S}^{0} w(p)S(p)dp. \tag{3}$$

Here the integration is over the pressure ($p$) coordinate from the surface ($p_S$) to the top of the atmosphere. The shape factor is

10 the normalized vertical profile of mixing ratio: $S(p)=C(p)\Omega_A/\Omega$ where $C$ is the HCHO mixing ratio and $\Omega_A$ is the total air column [Palmer et al., 2001]. The scattering weight measures the sensitivity of the backscattered radiation to the presence of HCHO at a given pressure.

All satellite data products (except OMPS-PCA) in Table 1 report for each retrieval $\Omega$, $AMF_G$, and $AMF$, and the scattering weights $w(p)$ or equivalent averaging kernels $A(p)=w(p)/AMF$. The BIRA retrievals report in addition the

15 corrected SCD $\Delta\Omega_S$ and background correction $\Omega_o$. To be able to interpret differences between retrievals, we obtained the $\Omega_o$ values used by the SAO retrievals and applied Equation (1) compute their values of $\Delta\Omega_S$. For OMPS-PCA, we computed the AMF based on the its reported shape factors $S(p)$, $w(p)$ and $AMF_G$ by Equation (3), computed $\Omega_o$ based on its reported uncorrected and corrected VCDs, and then obtained $\Delta\Omega_S$ by Equation (1).

## 3 Aircraft observations and GEOS-Chem model simulation

The SEAC[4]RS DC-8 aircraft flew 21 flights over the Southeast US between 5 August and 25 September 2013, providing extensive mapping of the mixed layer and vertical profiling from the mixed layer to the upper troposphere (Figure 1). The mixed layer is defined here as the convectively unstable region of the atmosphere in contact with the ground, as determined from the aircraft by aerosol lidar [Browell et al., 1989; Hair et al., 2008; DIAL-HSRL Mixed Layer Heights README, 2014; Scarino et al., 2014]. It typically extended to 1.5–2 km altitude during the afternoon. The mixed layer was

often capped by a convective cloud layer of fair-weather cumuli extending to about 3 km, with the free troposphere above [Kim et al., 2015]. Most flight hours were in the afternoon, and 95% between 0930 and 1800 local time for the data in Figure 1. Diurnal variability in the HCHO column is expected to be small since photochemistry is both a source and a sink [Millet et al., 2008; Valin et al., 2016].

Figure 2 (left panel) shows point-to-point comparison between 1-minute averaged ISAF and CAMS HCHO

observations (R3 version) aboard the aircraft. There is excellent correlation in the mixed layer ($r$=0.96) and above ($r$=0.99). Reduced major axis (RMA) regression of the two data sets yields a slope of 1.10±0.00, with ISAF 10% higher than CAMS.





The good agreement between CAMS and ISAF provides confidence that they can be used for satellite validation purposes. The aircraft data show high concentrations in the mixed layer due to biogenic isoprene emission, and a sharp drop above the mixed layer because of the short lifetimes of isoprene (~1 h) and of HCHO itself (~2 h).

Part of the horizontal variability observed in the mixed layer reflects local temperature (affecting isoprene emission)
and mixing depth at the time of the flights. We tried to remove these dependences in order to derive mean HCHO columns for the SEAC[4]RS period (5 August–25 September) that could be compared to the mean spatial patterns in the satellite data. Day-to-day variability in HCHO columns can be fitted well to $\exp[0.11T]$ where $T$ is the surface air temperature in K and the exponential dependence is that of isoprene emission [Palmer et al., 2006; Zhu et al., 2014]. We applied this temperature dependence to the aircraft HCHO mixed layer concentrations in Figure 1 in order to correct for the difference between the
local surface air temperature at the time of the flight and the local mean midday (1200–1300 local time) surface air temperature for the SEAC[4]RS period. Temperatures were taken from the Goddard Earth Observing System-Forward Processing (version 5.11.0, GEOS-FP hereafter) assimilated meteorological data product of the NASA Global Modeling and Assimilation Office (GMAO) [Molod et al., 2012]. We then converted these mean mixing ratios to HCHO columns by using the mean vertical profile information in Figure 1. This step assumed uniform HCHO mixing ratios from the surface up
through the local mixing depth measured from the aircraft [DIAL-HSRL Mixed Layer Heights README, 2014], an exponential decay from the top of the mixed layer to 650 hPa with a scale height of 1.9 km, and a fixed background of $0.40 \times 10^{16}$ molecules cm$^{-2}$ above.

The bottom left panel of Figure 1 shows the resulting mean HCHO columns for the SEAC[4]RS period, as inferred from the CAMS measurements. The spatial distribution is markedly different and smoother than for the original mixed layer
data (top left panel), reflecting in large part the normalization to mean local temperatures for the SEAC[4]RS period. Figure 3 shows the spatial distribution of midday temperatures for the SEAC[4]RS period, along with base isoprene emissions at 303 K from the MEGAN 2.1 model [Guenther et al., 2006; 2012]. The base isoprene emissions reveal a hotspot in the Ozarks region of Southeast Missouri (dense oak cover). This region was repeatedly sampled by the aircraft on hot days. The HCHO aircraft observations are particularly high there but this feature is muted after correction for the mean August–September
temperatures, which are much cooler in Missouri than further south. Inferred HCHO columns in Figure 1 are instead highest over Arkansas and Louisiana, where August–September temperatures are high.

We simulated the SEAC[4]RS period using the GEOS-Chem v9-02 CTM (http://geos-chem.org) with 0.25°×0.3125° horizontal resolution over North America driven by NASA GEOS-FP assimilated meteorological fields. The model has 47 vertical levels including 18 below 3 km. Initial simulations of the SEAC[4]RS data with GEOS-Chem pointed to a positive
bias in the daytime GEOS-FP diagnostic for the height of the mixed layer (mixing depth), used in GEOS-Chem for surface-driven vertical mixing. Comparisons of GEOS-FP mixing depths to lidar and ceilometer data for other field studies in the Southeast US found a 30–50% bias [Scarino et al., 2014; Millet et al., 2015]. For the SEAC[4]RS simulation we decreased the GEOS-FP mixing depths by 40%, and comparison to the aircraft lidar measurements along the DC-8 flight tracks shows that



this corrects the bias (Figure 4). Corrected afternoon (1200–1700 local time) GEOS-FP mixing depths along the flight tracks in the Southeast US average 1530±330 m, compared to 1690±440 m in the lidar data.

Formaldehyde production in GEOS-Chem over the Southeast US in summer is mainly from isoprene. Companion papers by Fisher et al. [2016], Marais et al. [2016], and Travis et al. [2016] describe the GEOS-Chem simulation of isoprene

chemistry in SEAC[4]RS and comparisons to aircraft and surface observations. Biogenic VOC emissions are from the MEGAN 2.1 model as implemented in GEOS-Chem by Hu et al. [2015] and with a 15% decrease applied to isoprene [Wolfe et al., 2015]. Surface-driven vertical mixing up to the mixing depth is as described by Lin and McElroy [2010].

Figure 2 (right panel) compares simulated and observed HCHO mixing ratios along the SEAC[4]RS flight tracks, averaged over the GEOS-Chem grid and time step. Comparison for the ensemble of data shows high correlation ($r$=0.80) and

no significant bias. Part of the correlation reflects the dependence on altitude, which is very well captured by GEOS-Chem (Figure 1, right panel). Removing this dependence on altitude, the correlation between model and observations remains high within the mixed layer ($r$=0.64) with only a small bias (-3±2%) indicated by the RMA linear regression. GEOS-Chem is less successful in reproducing the HCHO concentrations in the free troposphere, with a -41% normalized mean bias, which may due to insufficient deep convection in the model.

Integration of the mean vertical profiles in Figure 1 indicates a mean GEOS-Chem HCHO column of $1.46\times10^{16}$ molecules cm$^{-2}$, which is 10% lower than observed by CAMS ($1.63\times10^{16}$ molecules cm$^{-2}$), and 23% lower than observed by ISAF ($1.90\times10^{16}$ molecules cm$^{-2}$). The spatial correlation between GEOS-Chem mean HCHO columns (Figure 3, bottom panel) and the HCHO columns inferred from the CAMS data is 0.44 on the 0.5$^{o}$×0.5$^{o}$ grid, with GEOS-Chem capturing the region of maximum HCHO in Arkansas and Louisiana.

**4 Intercomparison and validation of satellite data sets**

Figure 5 shows the spatial distribution of mean HCHO columns over the SEAC[4]RS period taken from the six satellite retrievals of Table 1, along with values from GEOS-Chem and columns inferred from the CAMS aircraft observations. All retrievals feature maximum values over and around Arkansas and Louisiana, consistent with GEOS-Chem and CAMS and indicative of the maximum in isoprene emission (Figure 3). They are also consistent in showing high values

over the Southeast US due to isoprene emission.

Spatial correlations of HCHO columns on the 0.5$^{o}$×0.5$^{o}$ grid of Figure 5 are given in Table 2. Correlation coefficients for the different satellite retrievals are only 0.24–0.44 with CAMS but 0.38–0.85 with GEOS-Chem and typically 0.4–0.8 between pairs of retrievals. We conclude that there is consistency between retrievals in the spatial information even at the 0.5$^{o}$ scale. The GOME2A-BIRA retrieval is noisier than the others and we attribute this to

degradation of the instrument after 7 years of operations [De Smedt et al. 2015]. GOME2A has operated in reduced swath mode since July 2013, reducing its coverage by half and thus leading to greater noise in the time-averaged data





We see from Figure 5 that all retrievals are biased low relative to CAMS and GEOS-Chem. Table 3 gives statistics for these biases as spatial averages for the Southeast US. GEOS-Chem columns are sampled on the same schedule and scenes as the individual retrievals, and are increased by 10% to correct for the bias with CAMS. Satellite retrieval biases relative to the corrected GEOS-Chem values range from -20% (OMI-BIRA) to -51% (OMPS-PCA). The GOME2A and

GOME2B observations are made at 0930 local time, while the OMI and OMPS observations are made at 1330 local time. GEOS-Chem columns increase by 6% from 0930 to 1330 and this is accounted for in the GEOS-Chem comparisons of Table 3.

Retrieval biases in the vertical column $\Omega$ could be contributed by the corrected slant column ($\Delta\Omega_S$), the AMF, and the background correction $\Omega_o$ (Equation (1)). Table 3 gives mean values for these different terms. We see that the OMI-

BIRA column is the highest because it has the highest $\Delta\Omega_S$ and lowest AMF, while the OMPS-PCA column is the lowest because its $\Delta\Omega_S$ is the lowest. OMPS-SAO and OMPS-PCA use the same OMPS spectra but the OMPS-SAO $\Delta\Omega_S$ are much higher and more consistent with the other retrievals. One caveat is that the derived $\Delta\Omega_S$ of OMPS-PCA may not be the best measure for its algorithm sensitivity, since OMPS-PCA doesn't retrieve a slant column nor does it subtract the Pacific SCD to remove offsets, as described in section 2.

GOME2A-BIRA columns average 18% lower than GOME2B-BIRA despite sharing the same retrieval algorithm and overpass time. Although GOME2A was very fine for the first 5 years of operation (2007–2011) [De Smedt et al. 2012; 2015], it is noisier than the others during the SEAC[4]RS period, as pointed out above, reflecting instrument degradation and the lower number of observations.

The OMI-BIRA retrieval has the smallest bias relative to the GEOS-Chem and CAMS HCHO columns, and this is

due in part to its low AMF (0.88). Figure 6 shows the mean reported scattering weights and shape factors for that retrieval (Equation (3)), in comparison to other retrievals and to the CAMS aircraft observations. OMI-BIRA has lower scattering weights than the other retrievals, contributing to the lower AMF, and we discuss that below. The shape factor in the OMI-BIRA retrieval (from the IMAGES CTM with horizontal resolution of $2^o \times 2.5^o$) underestimates HCHO in the boundary layer and overestimates it in the free troposphere, most likely due to the use of IMAGES profiles for the year 2012, as IMAGES

profiles for the year 2013 were not available when the BIRA retrievals were processed. With the correct shape factor from CAMS the OMI-BIRA retrieval has an even lower AMF (0.74), as shown in Table 3, making it even better in comparison to GEOS-Chem and to the aircraft data.

Table 3 also gives the AMFs for the other retrievals re-computed using CAMS shape factors. The differences with the original AMFs are less than 6.0%, a much smaller correction than for OMI-BIRA. Although the results for OMI-BIRA

illustrate how sensitive the AMF calculation is to the specification of shape factor, we find that this is not a significant source of bias in the other retrievals. This may reflect compensating errors in the vertical profile, as illustrated in Figure 6 with the OMI-SAO shape factors in comparison with CAMS.



When the AMFs for all retrievals are re-computed with common CAMS shape factors, as shown in Table 3, the remaining differences in AMFs are solely driven by scattering weights and viewing angles. Figure 6 shows that scattering weights are 10–30% higher in the OMI-SAO retrieval (AMF=1.02) than in the OMI-BIRA retrieval (AMF=0.85). The difference remains for cloud-free satellite pixels (cloud fraction <0.01) and so is not due to different treatments of cloud

effects. Surface reflectivity averages 0.048 in OMI-SAO and 0.037 in OMI-BIRA. Although both use the OMI surface reflectance climatology of Kleipool et al. [2008], OMI-SAO applies monthly mean reflectivities while OMI-BIRA applies monthly minimum reflectivities. This would contribute in part to the difference in scattering weights at lower altitudes. The scattering weights of OMI-BIRA are lower than those of GOME2B-BIRA (Figure 6) and GOME2A-BIRA (about the same as GOME2B-BIRA, not shown), even though all BIRA retrievals use the same surface reflectivity. Although reduced, part of

the difference remains for cloud-free satellite pixels (cloud fraction <0.01).

The background corrections ($\Omega_o$=0.30–0.38×10$^{16}$ molecules cm$^{-2}$) in the different retrievals are all consistent and amount to about 30% of the mean columns $\Omega$ over the Southeast US. They agree with background HCHO columns measured by aircraft over the remote North Pacific (0.37±0.09×10$^{16}$ molecules cm$^{-2}$, Table 8 in Singh et al. [2009]).

Previous studies have shown that month-to-month variability in HCHO columns seen from space over the Southeast

US in summer is mainly driven by the temperature dependence of isoprene emission [Palmer et al., 2006; Millet et al., 2008; Duncan et al., 2009; Zhu et al., 2014]. Figure 7 shows time series of daily HCHO columns averaged spatially over the Southeast US for the six different retrievals. All retrievals have day-to-day temporal coherence consistent with the temperature dependence of isoprene emission. Temporal correlation between the daily HCHO column and midday temperature ranges from 0.52 to 0.75 for the different retrievals. GOME2A-BIRA shows the lowest correlation with

temperature, again likely due to noise from instrument degradation.

**5 Conclusions**

We have used SEAC$^4$RS aircraft observations of formaldehyde (HCHO) from two in situ instruments over the Southeast US for 5 August–25 September 2013, together with a GEOS-Chem chemical transport model simulation at 0.25$^o$×0.3125$^o$ horizontal resolution, to validate and intercompare six HCHO retrievals from four different satellite

instruments operational during that period. The combination of aircraft data and GEOS-Chem model fields provides strong constraints on the mean HCHO columns and their variability over the Southeast US, with high column amounts driven by biogenic isoprene emission.

We find that all retrievals show a large degree of consistency in their simulation of spatial and temporal variability. All retrievals capture the HCHO maximum over Arkansas and Louisiana seen in the aircraft data and in GEOS-Chem, and

corresponding to the region of highest isoprene emission. Spatial correlation coefficients between retrievals are relatively high (0.4–0.8) even on a 0.5$^o$×0.5$^o$ grid. All retrievals are also consistent in their simulation of day-to-day variability correlated with temperature. This success demonstrates that HCHO columns observed from space can provide a reliable





proxy for isoprene emission. GOME2A-BIRA (launched in 2006) is noisier than other retrievals. We attribute this to instrument degradation and reduced sampling.

Despite this success and consistency in observing HCHO variability from space, we find that all satellite retrievals are biased low in the mean, by 20% to 51% depending on the retrieval. This would cause a corresponding bias in estimates of isoprene emission made from the satellite data. The bias is smallest for OMI-BIRA and could be further reduced by correcting the assumed HCHO vertical profiles (shape factors) assumed in the AMF calculation. Other retrievals have larger biases that appear to reflect a combination of (1) spectral fitting affecting the corrected slant columns, and (2) scattering weights in the radiative transfer model (affecting the AMF). Aside from OMI-BIRA, the shape factors used in the retrievals are not a significant source of error.

Our work points to the need for improvement in satellite HCHO retrievals to correct the mean low bias. Focus should be on slant column fitting, on corrected slant columns, and on the calculation of scattering weights. OMI-BIRA has the largest corrected slant columns and smallest scattering weights of all retrievals, yielding the best match to the SEAC[4]RS observations. This can provide a comparison reference for other retrievals.

**Acknowledgments**

We acknowledge contributions from the NASA SEAC[4]RS Science Team, especially the CAMS, ISAF, and DIAL/HSRL lidar group. We would also like to thank the SEAC[4]RS flight crews and support staff for their outstanding efforts in the field. This work was funded by the US National Aeronautics and Space Administration. We thank Michel Van Roozendael for helpful discussions. Jenny A. Fisher acknowledges support from a University of Wollongong Vice Chancellor's Postdoctoral Fellowship.



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





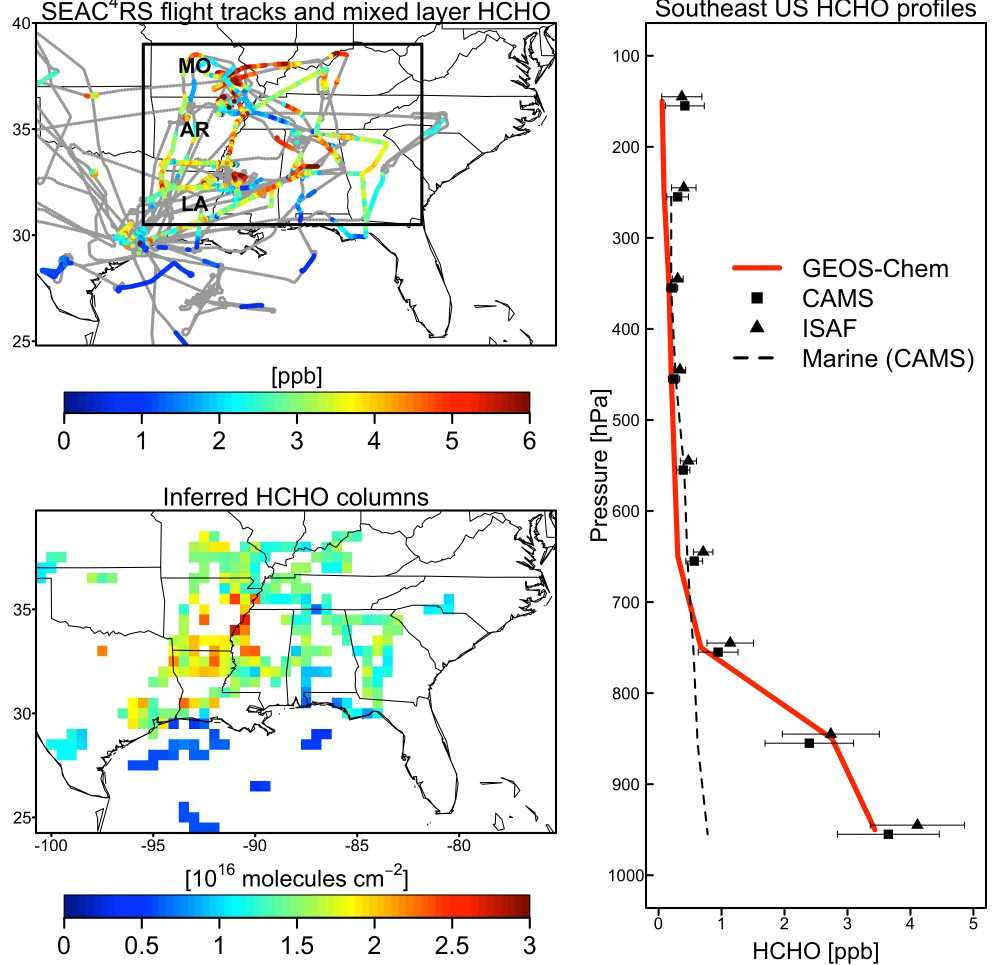

**Figure 1:** Formaldehyde (HCHO) concentrations along SEAC⁴RS aircraft flight tracks (5 August–25 September 2013). The top left panel shows the DC-8 flight tracks (in grey) and the CAMS measurements aboard the aircraft in the mixed layer. The mixed layer is the convectively unstable region of the atmosphere in contact with the surface, diagnosed locally from aerosol lidar observations aboard the
5   aircraft [DIAL-HSRL Mixed Layer Heights README, 2014] and typically extending to 1–3 km altitude. The states of Missouri (MO), Arkansas (AR), and Louisiana (LA) are indicated. The right panel shows the mean vertical profiles observed by the CAMS and ISAF instruments, and simulated by GEOS-Chem, for the Southeast US domain (30.5°–39°N, 95°–81.5°W) defined by the black rectangle in the top left panel. Horizontal bars represent observed standard deviations. GEOS-Chem is sampled along the flight tracks at the time of the measurements. The dashed black line shows the mean vertical CAMS profile in marine air over the Gulf of Mexico (22°–28°N, 96.5°–
10  88.5°W). The bottom left panel shows the mean HCHO columns on a 0.5°×0.5° grid derived from the CAMS measurements after normalizing for temperature, for mixed layer depth, and for the contribution from HCHO aloft (see text).





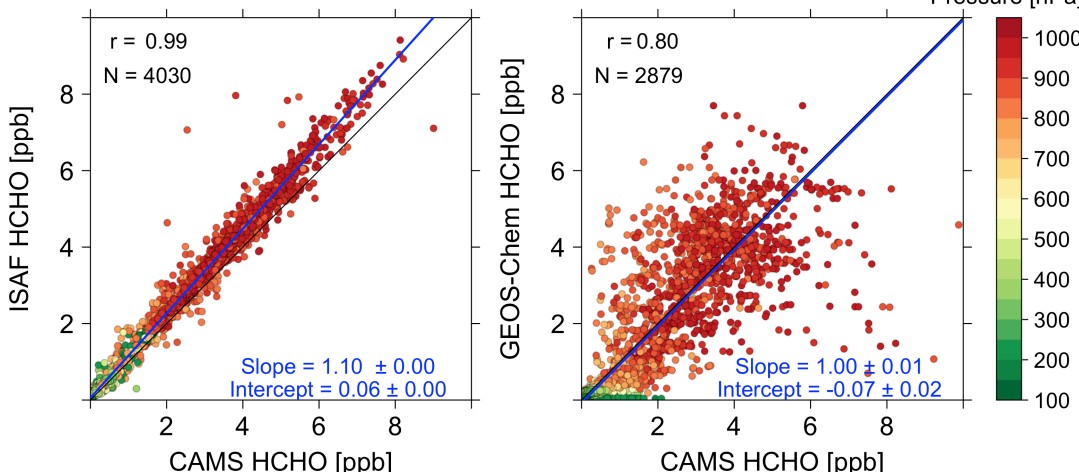

**Figure 2**. Comparisons between HCHO measurements from the CAMS and ISAF instruments aboard the SEAC⁴RS aircraft, and simulated by GEOS-Chem, for the Southeast US flight tracks (box in Figure 1). The left panel compares 1-minute measurements from CAMS and ISAF. The right panel compares GEOS-Chem and CAMS HCHO. Here and elsewhere for model-observation comparisons, HCHO observations along the flight tracks are averaged onto the GEOS-Chem grids (0.25°×0.3125°, 47 vertical layers) and time steps (10 minutes). HCHO data points are colored by atmospheric pressure. Slopes and intercepts of reduced major axis (RMA) regressions are shown along with the correlation coefficient (*r*), sample size (*N*), RMA regression line (in blue), and 1:1 line.





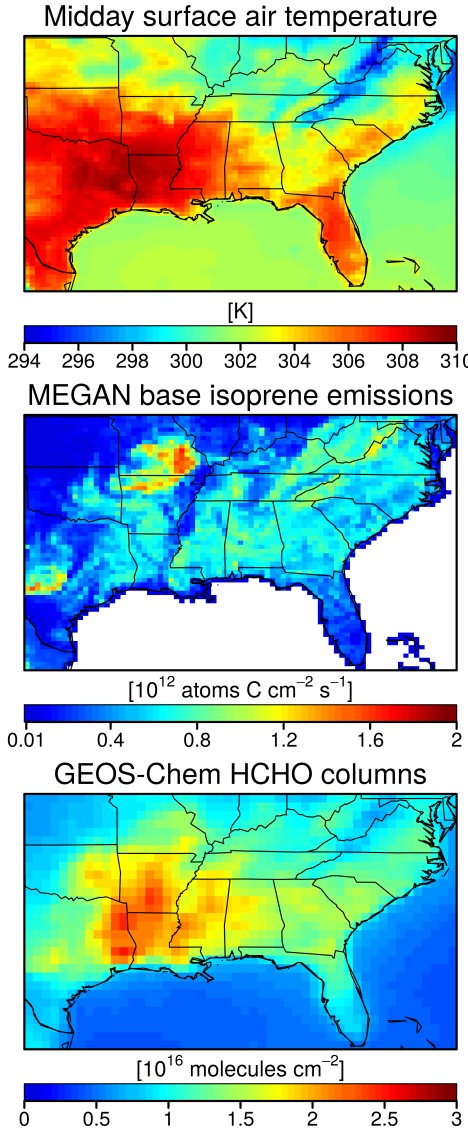

**Figure 3.** Mean temperature, base isoprene emissions, and HCHO columns in the GEOS-Chem model for the SEAC[4]RS period (5 August–25 September 2013). The top panel shows the midday (1200–1300 local time) surface air temperature from the GEOS-FP assimilated meteorological data. The middle panel shows the MEGAN 2.1 base isoprene emissions from Guenther et al. [2006; 2012] for standard conditions (air temperature=303 K, photosynthetic photon flux density=200 μmol m$^{-2}$ s$^{-1}$ for sunlit leaves and 50 μmol m$^{-2}$ s$^{-1}$ for shaded leaves.) The bottom panel shows the GEOS-Chem HCHO columns computed with MEGAN 2.1 isoprene emissions and sampled at 1330 local time (OMI schedule).





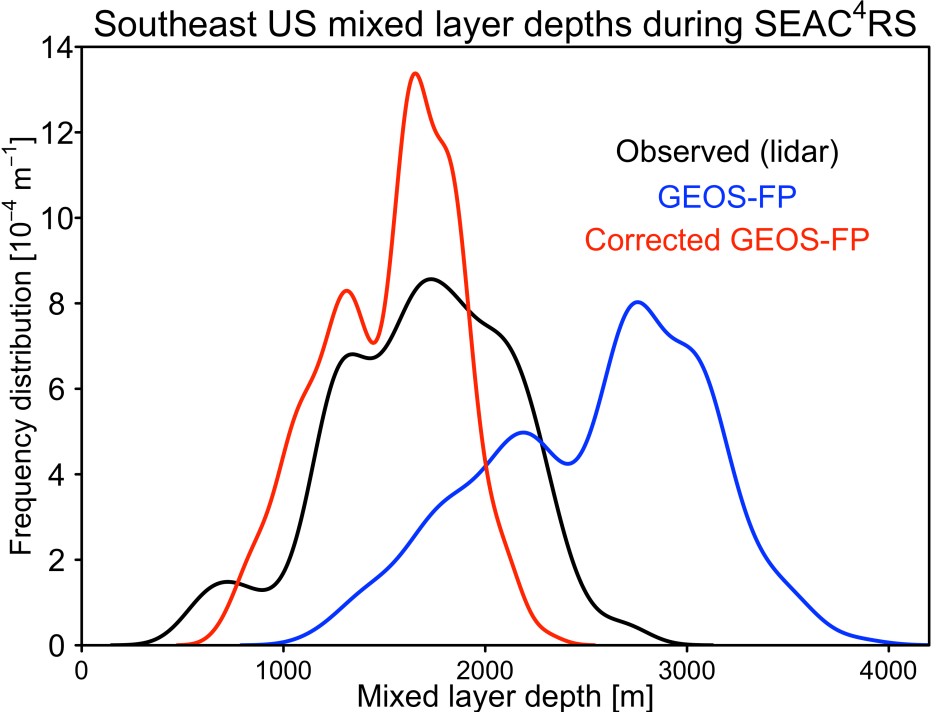

**Figure 4.** Frequency distribution of mixed layer depths over the Southeast US during the SEAC[4]RS period (5 August–25 September 2013). Observations by aerosol lidar aboard the aircraft [DIAL-HSRL Mixed Layer Heights README, 2014] are compared to the local GEOS-FP data used to drive GEOS-Chem, before and after the 40% downward correction. The frequency distributions are constructed from 1-minute average data along the aircraft flight tracks over the Southeast US (box in Figure 1) for the 1200–1700 local time window.





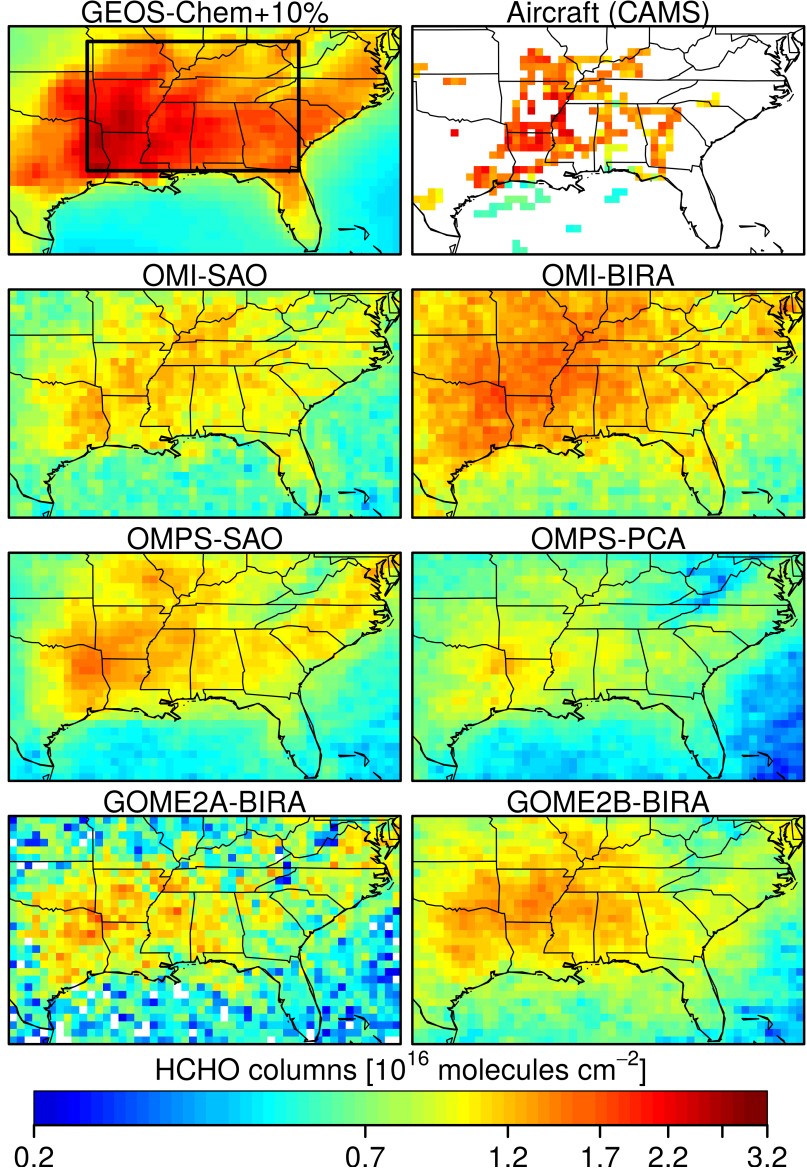

**Figure 5.** HCHO vertical column densities over the Southeast US averaged over the SEAC[4]RS period (5 August–25 September 2013). The bottom panels show six retrievals from four satellites (OMI, GOME-2A, GOME-2B and OMPS) and three different groups (Table 1). The top panels show (1) GEOS-Chem model results sampled on the OMI schedule and increased by 10% to correct for the bias relative to CAMS aircraft measurements; and (2) columns derived from the CAMS aircraft measurements (same as bottom left panel of Figure 1 but on a different color scale). The black rectangle represents the Southeast US domain (same as in Figure 1). Color bar is a logarithmic scale.




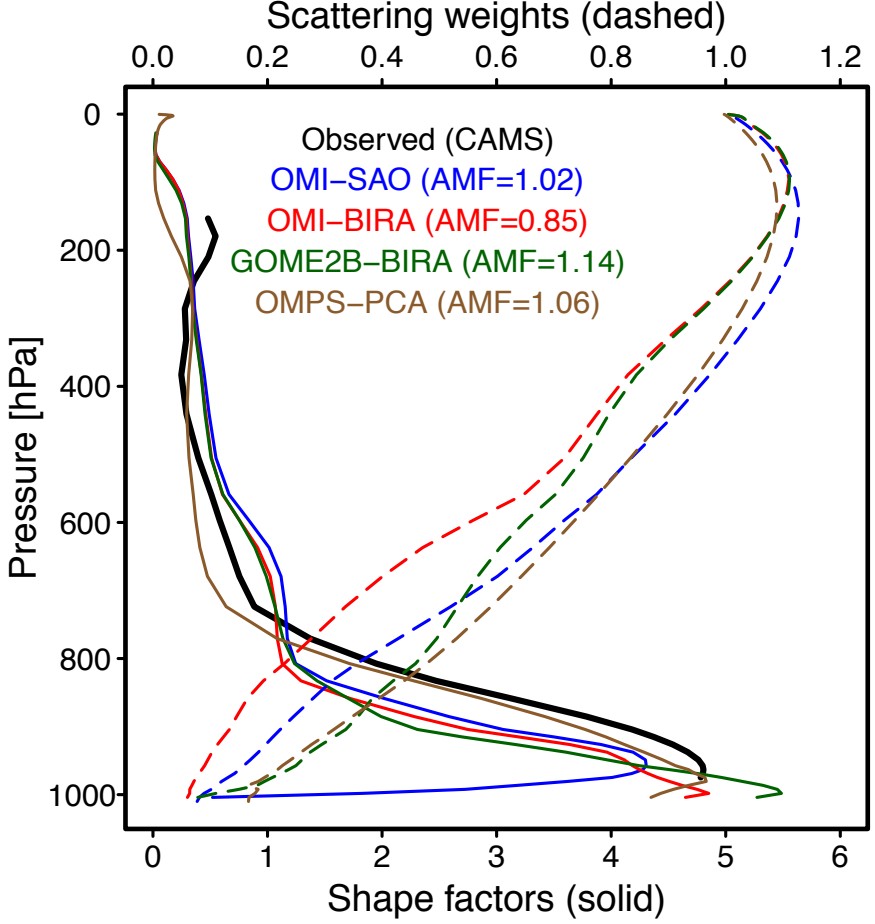

**Figure 6.** Mean scattering weights ($w$) and shape factors ($S$) for HCHO retrievals over the Southeast US during the SEAC[4]RS period, and resulting air mass factor (AMF) computed from equation (3). Values are shown for the OMI-SAO, OMI-BIRA, GOME2B-BIRA, and OMPS-PCA retrievals. Also shown are the observed HCHO shape factors (black) from the mean CAMS profile in Figure 1.





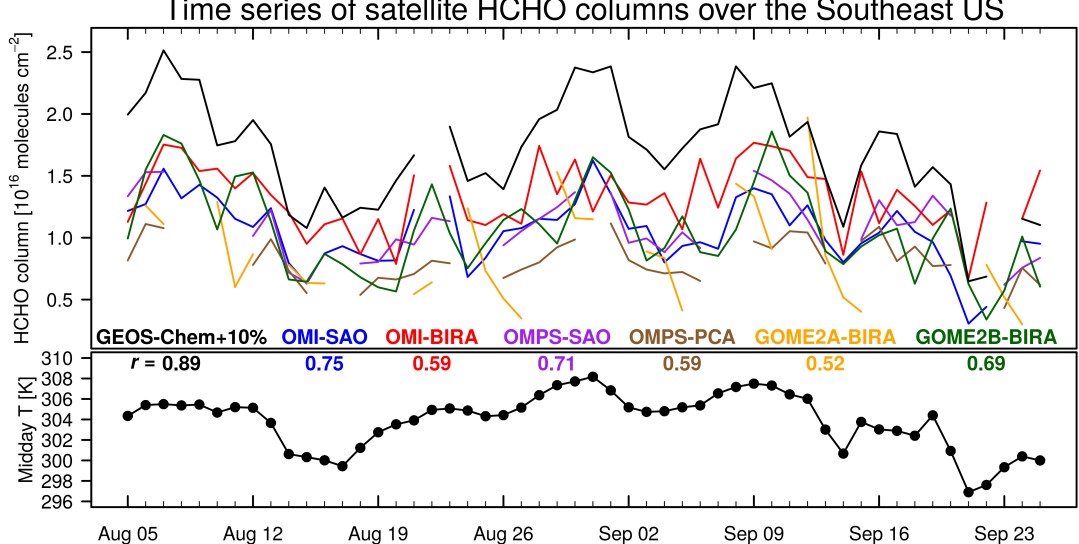

**Figure 7.** Daily variability of HCHO vertical column densities over the Southeast US during SEAC$^4$RS. The top panel shows daily HCHO columns averaged over the Southeast US (box in Figure 5) for the different satellite retrievals of Table 1. GEOS-Chem columns (black) are sampled following the OMI viewing geometry and scaled up by 10% on the basis of comparison with CAMS aircraft columns. The bottom panel shows the local midday (1200–1300 local time) surface air temperature over the Southeast US domain from the GEOS-FP assimilated meteorological data. Also shown for each data set is the temporal correlation coefficient ($r$) with temperature.





**Table 1.** Satellite HCHO products validated and intercompared in this work[a]

| Retrieval | Nadir resolution [km²] | Local viewing time (LT) | Chemical transport model[b] | Detection limit[c] [$10^{16}$ molecules cm$^{-2}$] | Reference[d] |
|---|---|---|---|---|---|
| OMI-SAO (V003) | 24×13 | 1330 | GEOS-Chem v09-01-03 | 1.0 | [1] |
| OMI-BIRA (V14) | 24×13 | 1330 | IMAGES v2 | 0.7 | [2] |
| GOME2A-BIRA (V14) | 40×40 | 0930 | IMAGES v2 | 0.8 | [3] |
| GOME2B-BIRA (V14) | 80×40 | 0930 | IMAGES v2 | 0.5 | [2] |
| OMPS-SAO | 50×50 | 1330 | GEOS-Chem v09-01-03 | 0.75 | [4] |
| OMPS-PCA | 50×50 | 1330 | GMI | 1.2 | [5] |

[a] Retrievals operational during the SEAC⁴RS aircraft campaign (5 August–25 September 2013). These include retrievals from four different sensors (OMI, GOME-2A, GOME-2B and OMPS), flown on different platforms, with different retrievals for OMI and OMPS produced by the Harvard Smithsonian Astrophysical Observatory (SAO), the Belgian Institute for Space Aeronomy (BIRA), and the NASA Goddard Space Flight Center by Principal Component Analysis (PCA).

[b] Chemical transport model (CTM) supplying the normalized mixing ratio vertical profiles (shape factors) and background correction ($\Omega_o$ see section 2) used in the retrieval. References are Miller et al. [2014] for GEOS-Chem v09-01-03, Stavrakou et al. [2009] for IMAGES v2, and Rodriguez [1996] for GMI.

[c] Single-scene spectral fitting uncertainties (except for OMPS-PCA). Fitting windows for the different retrievals are in the 327.7–356.5 nm range. Detection limits of GOME2A and B were in 2013. Detection limit of OMPS-PCA is determined as the 4-times of 1-sigma noise over the Pacific Ocean [Li et al., 2015].

[d] [1] Gonzalez Abad et al. [2015a]; [2] De Smedt et al. [2015]; [3] De Smedt et al. [2012]; [4] Gonzalez Abad et al. [2015b]; [5] Li et al. [2015]. OMI-SAO data were downloaded from: http://disc.sci.gsfc.nasa.gov/Aura/data-holdings/OMI/omhcho_v003.shtml. GOME2A-BIRA and GOME2B-BIRA data were downloaded from: http://h2co.aeronomie.be. Other data were courtesy of the retrieval group.



**Table 2**. Spatial/temporal correlation coefficients ($r$) between pairs of HCHO column products[a]

| HCHO product | OMI-SAO (V003) | OMI-BIRA | GOME2A-BIRA (V14) | GOME2B-BIRA (V14) | OMPS-SAO | OMPS-PCA |
|---|---|---|---|---|---|---|
| OMI-SAO (V003) | 1/1 | | | | | |
| OMI-BIRA | 0.55/0.67 | 1/1 | | | | |
| GOME2A-BIRA (V14) | 0.28/0.48 | 0.38/0.50 | 1/1 | | | |
| GOME2B-BIRA (V14) | 0.50/0.76 | 0.65/0.60 | 0.49/0.26 | 1/1 | | |
| OMPS-SAO | 0.48/0.77 | 0.70/0.50 | 0.45/0.55 | 0.72/0.76 | 1/1 | |
| OMPS-PCA | 0.40/0.70 | 0.60/0.51 | 0.53/0.63 | 0.71/0.68 | 0.85/0.84 | 1/1 |
| GEOS-Chem[b] | 0.38/0.88 | 0.50/0.65 | 0.68/0.82 | 0.85/0.88 | 0.74/0.86 | 0.82/0.75 |
| Aircraft (CAMS)[c] | 0.24/- | 0.44/- | 0.26/- | 0.35/- | 0.43/- | 0.37/- |

[a] Correlation coefficients between HCHO columns for different pairs of satellite retrievals, GEOS-Chem, and CAMS aircraft observations. Values are for the Southeast US domain (box in Figure 1 and 5) during SEAC[4]RS (5 August–25 September 2013). Spatial correlation coefficients are computed for the temporally averaged data on the 0.5º×0.5º grid of Figure 5. Temporal correlation coefficients are computed from daily averages of each retrieval over the Southeast US domain.

[b] GEOS-Chem CTM columns sampled for the same scenes as the individual retrievals.

[c] Aircraft column data are temporal averages for the SEAC[4]RS period as shown in Figure 1 (bottom left panel) and Figure 5 (top right panel).



**Table 3.** Satellite retrievals of HCHO columns over the Southeast US[a]

| Retrieval | Mean values[b] | | | | | | with CAMS shape factors | | | GEOS-Chem+10%[c] |
|---|---|---|---|---|---|---|---|---|---|---|
| | $\Omega$ | $\Delta\Omega_S$ | $AMF_G$ | $AMF$ | $\Omega_o$ | Bias[d] | $AMF$[e] | $\Omega$[f] | Bias[d] | $\Omega$ |
| OMI-SAO (V003) | 1.06 | 0.65 | 2.66 | 0.95 | 0.38 | -37% | 1.01 | 0.96 | -43% | 1.69 |
| OMI-BIRA | 1.33 | 0.87 | 2.62 | 0.88 | 0.31 | -20% | 0.74 | 1.47 | -12% | 1.67 |
| GOME2A-BIRA (V14) | 0.89 | 0.62 | 2.37 | 1.12 | 0.30 | -44% | 1.14 | 0.84 | -47% | 1.59 |
| GOME2B-BIRA (V14) | 1.09 | 0.86 | 2.56 | 1.22 | 0.30 | -31% | 1.27 | 0.98 | -38% | 1.59 |
| OMPS-SAO | 1.09 | 0.72 | 2.54 | 1.01 | 0.38 | -34% | 1.02 | 1.01 | -39% | 1.66 |
| OMPS-PCA | 0.80 | 0.49 | 2.53 | 1.11 | 0.35 | -51% | 1.15 | 0.78 | -52% | 1.63 |

[a] Mean values over the Southeast US domain (box in Figure 1 and 5) for the data in Figure 5 collected during the SEAC[4]RS period (5 August–25 September, 2013).

[b] Mean values provided as part of the operational and research retrieval product including vertical HCHO columns ($\Omega$), corrected slant
5   columns ($\Delta\Omega_S$), geometrical AMF ($AMF_G$), air mass factors ($AMF$), and background correction ($\Omega_o$), following equation (1). Columns are in units of [$10^{16}$ molecules cm$^{-2}$] and AMFs are dimensionless. The corrected slant columns and background correction are not reported in the SAO and OMPS-PCA retrievals and are reconstructed here to enable comparison with the other retrievals (see section 2).

[c] GEOS-Chem columns sampled for the same scenes as the individual retrievals and increased by 10% to correct for the bias relative to the SEAC[4]RS aircraft measurements (Figure 1). Mean GEOS-Chem columns increase with time of day by 6.0% from 0930 local time
10   (GOME2A and B) to 1330 (OMI and OMPS).

[d] Normalized mean bias relative to the corrected GEOS-Chem values (last column in the Table).

[e] AMFs recalculated using the mean HCHO vertical shape factor from the CAMS aircraft instrument (Figure 1 and 6) and the scattering weights or averaging kernels provided as part of the satellite product (Figure 6).

[f] Columns recomputed using AMFs constrained by the CAMS aircraft measurements.