# Peer review of "Observing atmospheric formaldehyde (HCHO) from space: validation and intercomparison of six retrievals from four satellites (OMI, GOME2A, GOME2B, OMPS) with SEAC4RS aircraft observations over the Southeast US"

_Atmospheric Chemistry and Physics, 2016_

## Referee Comment (RC1) · Anonymous Referee #1 · 5 Apr 2016

This is a long over due paper concerning the validation of satellite HCHO columns, retrieved by various sensors/groups data against in-situ aircraft measurements. It addresses an important topic and should be published when the following issues have been addressed. The paper is generally well written with nice graphics (apart from fig 7).

Although the analysis is decent, and it gives the reader an impression of the quality of the satellite HCHO column data, I do, however, feel that a somewhat opportunity has

been missed since the aircraft data could have been used to really pick the retrievals apart. Maybe this is planned for the future, one can only hope!

Specific Comments:

Page 2, Line 25 & subsequent paragraph: Please note that Barkley et al (2013) compared OMI data to GABRIEL aircraft HCHO measurements and found OMI was about 40% too low.

Table 1 – I would include much more information about the details of the retrieval and AMF algorithms, e.g., fitting windows, absorbers fitted, use of radiance references, CTM vertical and horizontal resolutions, profile (monthly vs daily), resolution of scattering weight grids, treatment of clouds, terrain corrections, treatment of aerosols etc. Also discuss explcilty exactly how the reference sector correction is done in each case – yes they all follow the same approach - but there are subtle differences. This is a validation paper after all, so the reader needs to be well informed about the retrieval differences.

Page 3, Line 21: You have gridded the data yourself, so why quote the error on the monthly means from De Smedt's 2008 paper? Surely you can compute this error for each retrieval.

Equation (1) - please state if sigma_0 is a vertical (I assume it is) or a slant column background.

Reference sector correction – since the retrieval groups are authors on the paper, I would urge the satellite data producers to provided their uncorrected slant columns (i.e. the raw retrieved slant columns) so that a consistent a reference sector adjustment could be applied to all. This really would add substantial value to the paper. But I suspect that it will not be attempted, which is a shame.

Page 4, line 27: "diurnal variability in the HCHO column is expected to be small..." Please quantify small – is this statement based on measurements or model studies?

Page 5: The normalisation process concerns me. I can sort of see why it was done but there are some opened questions. For example, what are the spatial gradients in mixing layer depths, how does this impact the HCHO column. It is stated that day to day variability is "fitted well" by exp[0.11T] – please quantify well? What is the uncertainty in the comparison procedure due to fitting this function? How does the satellite comparisons change say if an arbitrary function of exp(cT) is used instead, where 'c' is a different constant? Isoprene emissions, and thus HCHO columns, are very sensitive to temperature. What do you mean by 'local surface air temperature', where is this from (is it the GEOS-FP data) & how accurate is it? The conversion to a 'mean HCHO columns', using the mean vertical profile, again what is the error in this procedure and how does it impact the subsequent comparisons? What if you use a different assumed profile shape? Lastly, please discuss why it's really needed at all? Surely, you could compute HCHO columns for each flight, corresponding to a specific day and underlying surface footprint, and compare this to the satellite observations that lie in the same footprint (in which case you don't need to adjust for day-to-day variability)? I am, perhaps, being quite harsh but this step in the study is very important and yet it is given only a paragraph. Given the importance of the paper, I urge the authors to thoroughly discuss this key part of the analysis.

Section 4: Why don't the retrievals apply the same GEOS-Chem model background correction, and probably better still, compute the AMFs using the GEOS-Chem model data? This would help isolate the columns differences due to the spectral fitting approaches. Yes, this would be some work but it would really add another dimension to the paper. You're already half way there, using the mean CAMS profile anyway.

Page 8, lines 1-2: the AMFs are also sensitive to the respective cloud parameters for each retrieval (which instrument specific and subject to their own uncertainties), hence it is not just the scattering weights and angles.

Page 8, line 5 onwards: Again using the same surface reflectance data set in the AMF for all retrievals, would give some idea of its relative importance in this validation

exercise. Ok, each retrieval 'should' use a reflectance data set consistent with the cloud algorithm but it would be an interesting experiment to carry out.

Page 8, line 15: Am I missing something here, but you've corrected the GEOS-Chem model using the CAMS data, which is adjusted for day-to-day variability, and yet this shows 'good agreement' with temporal dependence of isoprene emissions. Isn't this a somewhat circular comparison/argument? Please elaborate, and if not, then make sure you adjust the text to avoid leaving the reader with this impression.

Page 8, line 30: I would say spatial correlations are 'moderate to relatively high', rather than give the feeling that the retrievals are better than they really are.

Page 9, final paragraph: yes you are right in what you say! But couldn't the authors address some of these issues in the framework of this paper?

Figure 3: are the temperature and HCHO columns, both sampled according to OMI observations (i.e. under the same cloud conditions?)

Figure 5, the GEOS-Chem data is sampled along the 'OMI schedule' – according to which OMI retrieval? I suspect that each retrieval will have different quality flags for their measurements, so a 'good' measurement in one might be flagged as 'bad' in another. Are the actual same OMI pixels gridded between the retrievals? This also holds for the OMPS data as well. Maybe discuss this in the main text to clarify.

Figure 6: why not show the AMFs alongside????

I'm sorry but Figure 7 looks like coloured spaghetti. Maybe make it multi-panelled, with GEOS-Chem plus OMI data, them GEOS-Chem plus the OMPS data etc.

Table 1 please explicitly state how the detection limits are calculated for each retrieval.

---

## Referee Comment (RC2) · Anonymous Referee #2 · 7 Apr 2016

General Comments This paper presents intercomparison results for HCHO vertical columns measurements from 4 satellite instruments and 6 different retrievals and comparison to an aircraft campaign over the Southeast US. The paper is easy to read and very interesting for the community in order to have a better view of the quality of the existing satellites HCHO retrievals. It is within the scope of ACP and should be published after taking into account the comments and improvements listed below.

Specific Comments page 3, line 3: it's the only point in the paper where the in-situ

data (reference and technique) are introduced, and as the whole paper is based on these measurements, it would be good to say a little more about them (maybe at the beginning of Section 3). What is their accuracy, sensitivity, ... see comment relative to page 4.

page 3, line 10: to my view, Table 1 should contain more information on the different retrievals. It should highlight the main differences between the various retrievals made on the same satellite sensor. Other differences than only the CTM are important: the albedo, surface, cloud correction, etcc (that are only mentioned in page 8) that impact AMF calculation, but also differences in DOAS settings for the slant column retrieval.

page 4, line 16: and applied Equation (1) "to" compute ... page 4, line 17: AMF based on the its reported: remove "the" or "its".

page 4, section 3: I would add a small paragraph on the 2 insitu techniques that are mentioned in the introduction, and that are considered the "truth" but that have 10% difference, as stated in line 31. Any hint on why? The organization of the figure sequence in this section is a little bit confusing for me (the different part of each figures are not discussed/mentioned together, but at different moments of section 3): first figure 1 (but only right and upper left panels), then Figure 2 (left panel), then bottom left panel of figure 1 (p.5, line 18), then figure 3 (p.5, line 20), then missing reference to figure 4 (p.5, line 29) and then only in page 6, the right panel of figure 2 is presented/mentioned. Maybe the bottom left panel of figure 1 could go with figure 3 and the right panel of figure 2 could go after figure 4?

page 4, line 24: it is stated that the typical extension of the mixed layer is to 1.5-2 Km in the afternoon, but in figure 1, the caption mention 1 to 3km. It would be good to add the height scale in km on the profile picture (and not only pressure) in order to easily make the link with the figure.

page 4, line 26: most flight hours were in the afternoon. How much is "most"? page 4, line 27: the diurnal variability is expected to be small. How much is "small"?

page 5, line 5: considering the 10% difference between the 2 in-situ technique, why is CAMS considered as the reference in the rest of the paper (GEOS-CHEM is increased by 10% to reach CAMS values) and not ISAF or the average of both?

page 5, line 29: add "as can be seen in figure 4, " before "Initial simulations ..." page 5, line 31: add "previous"before "comparisons of GEOS-FP ..."

page 6, line 1: add "red line in" before "figure 4". page 6, line 11: how do you "remove this dependence on altitude"? again, it would be helpful for an easy reading of the paper, to mention the limits of the mixed layer and of the free troposphere in pressure values. page 6, line 18: why only mention the spatial correlation with CAMS data and not with ISAF? page 6, line 26 and lower and Table 2: the spatial (and temporal) correlation coefficient calculation should be mentioned in the text here, and not only in the caption of Table 2.

page 6, line 29: GOME-2A BIRA is noisier wrt GOME-2B, this is clear. How much is this due do degradation and how much to the reduced swath mode?

page 7, line 2: "GEOS-Chem columns are sampled on the same schedule and scenes as the individual retrievals" but as GOME2A and GOME2B does not have the same swath, the pixels geometry and cloud impact are not the same, so why the value of GEOS-Chem columns in table 3 is the same (1.59) for both instruments?

page 7, line 8 to 14: This would be a good place to discuss the different retrieval choices and their impacts on the different retrieval steps (corrected slant column, AMF and background correction).

page 7, line 22: after discussing the OMI-BIRA shape factors (ie the IMAGES input) wrt to CAMS profile, you could discuss the other model shapes too? and how are GEOS-CHEM and IFAS profile? is this difference between model expected/known? what is the possible difference explanation?

page 7, line 29: the "much smaller" affirmation in the sentence "The differences with

the original AMFs are less than 6.0%, a much smaller correction than for OMI-BIRA" would be more clear with "are less than 0.06, while for OMI-BIRA the change in AMF is around the double (-0.14)".

page 8, line 2: not only "driven by scattering weights and viewing angles". The clouds, albedo, aerosols also impacts the AMFs. page 8, line 5: the difference in surface reflectivity between OMI-SAO and OMI-BIRA is not so small (∼23 to 29%). this certinaly impact (and not only "would contribute in part) "the difference in scattering weights ar lower altitudes". Please refer to sensitivity tests and AMF error estimations from literature.

page 8, line 8: the sentence "The scattering weights of OMI-BIRA are lower than those of GOME2B-BIRA (Figure 6) and GOME2A-BIRA (about the same as GOME2B-BIRA, not shown), even though all BIRA retrievals use the same surface reflectivity" is misleading. The retrieval is the same (same surface reflectivity), but the geometry of the 2 instruments is different, as well as the time of the day and clouds", so we expect different scattering weights.

page 8, line 14: a discussion of table 3 wrt to differences in the slant columns values between the different retrieval is missing (discussion on VCD, AMF and background correction has been done before).

page 8, line 18: what is the difference between the temporal correlation reported in figure 7 (and here in the text) and temporal correlation values of table 2 ?.

page 9, line 9: I find the sentence "Aside from OMI-BIRA, the shape factors used in the retrievals are not a significant source of error" a little bit odd. If I understand well table 3, the difference in VCD bias when considering the original profile and when using the CAMS profile is 8% (from -20% to -12% bias) for OMI-BIRA, which is indeed the highest change, but the other products ranges between 7% (for GOME2B-BIRA), to 6% (OMI-SAO), 5% (OMPS-SAO), 3% (GOME2A-BIRA) and finally 1% (OMPS-PCA), which are not so different than 8%. Please reformulate this sentence.

---

## Referee Comment (RC3) · Anonymous Referee #3 · 20 Apr 2016

This manuscript presents an intercomparison exercise of tropospheric HCHO retrieved from satellites measurements using several products obtained from independent retrieval approaches. In order to validate the products, the authors use aircraft observations during the short term field deployment of SEAC4RS in the Southeast of the US. This is an important validation effort. In general, the paper is well written and within the scope of ACP. That said, I have important comments/suggestion that I recommend revisions before final publication.

Major comments:

- In the abstract and conclusion sections it is mentioned that the HCHO columns are biased low by 20-50%, which is a significant number. Then, it is concluded that HCHO from satellite provides a reliable proxy for isoprene emissions. How do the authors conclude that HCHO is reliable given the significant bias (plus any uncertainty associated to the assumptions of HCHO yields from isoprene conversion)?. The only section with results shown is section 4, I recommend to insert a new section(s) where specif details are given regarding errors and how they affect the isoprene emission.

- The emissions of natural biogenic volatile organic compound have a temperature dependence, hence a seasonal variation that it is not studied in the present study. However, the statements along the manuscript are quite general regarding spatial and temporal distribution of HCHO. The focus of the study is only limited in the southeast of the US during less than two months. I suggest to state clearly that the results/conclusions shown are for this specific time/area.

- The aerosol optical depth (AOD) observed in the southeast of the US is significantly higher than other parts in north America. In addition, the temporal distribution of AOD over the southeast of the US has a maximum peak in the summer months (similar months as this study). However, it is not well documented in section 2 how aerosols are treated in the satellite retrievals. Also, current investigation in the southeast of the US associate aerosol aloft (e.g., Goldstein et al., 2008). I encourage the authors to explain in better detail the effect of aerosols in the retrieval and final uncertainty. How do aerosol aloft impact the retrieval of HCHO columns?.

- The in-situ sensors are taken as the ground truth in this study, however there is quite a bit of manipulation in the conversion of mixing ratios to columns. It is not clear to me why the HCHO need to be normalized if the columns are compared with satellite retrievals. Please give a thorough description of why this is needed and why the "Day-to-day variability in HCHO columns can be fitted well to exp[0.11T]". Is there

any explanation of why CAMS and ISAF have differences of about 10% since they measure the same air mass?.

- Have the authors looked at the trace gas inhomogeneities captured by the in-situ observations and compare with results of satellite on a pixel-pixel resolution and ?. In other words, do the correlation improve if the air mass sampled is the same?

- In the abstract it is mentioned that "The GEOS-Chem chemical transport model provides a common intercomparison platform". However, it is not clear how GEOS-Chem is used. Consider expanding the sentence explaining how this is achieved.

- Page 3, line 23: "Here and elsewhere, we use only satellite pixels with solar zenith angle less than 60, cloud fraction less than 0.3, and row anomalies (for OMI) screened". In the same paragraph it is mentioned the detection limit of the satellites but it is not clear if you use only data above the detection limit for the analysis in this work.

- Page 4, line 16: There are several abbreviations that need to be defined, e.g., SAO, OMPS-PCA, etc. A table/appendix with abbreviations would be useful.

- Page 5, line 5-17: The conversion of mixing ratios to columns is achieved by assuming that HCHO is co-located with aerosols (identified with the mixing height from DIAL-HSRL), how valid is this assumption?. I suggest to explain also why an exponential decay with a scale height of 1.9 is used and how do the background column is found.

---

## Author Comment (AC1) · 7 Jul 2016

We thank the three reviewers for their thorough and thoughtful comments.

Please see the attached zip file for our responses, the revised manuscript and the supplemental materials.

Please also note the supplement to this comment:

[Figure]

http://www.atmos-chem-phys-discuss.net/acp-2016-162/acp-2016-162-AC1-supplement.zip

---

## Author Comment (AC2) · 7 Jul 2016

We thank the three reviewers for their thorough and thoughtful comments. Please see the zip file above (under AC1) for our responses, the revised manuscript and the supplemental materials.

---

## Author Response (AR1)

We thank the three reviewers for their thorough and thoughtful comments. Our responses to reviewers are in blue.

**Anonymous Referee #1**

This is a long over due paper concerning the validation of satellite HCHO columns, retrieved by various sensors/groups data against in-situ aircraft measurements. It addresses an important topic and should be published when the following issues have been addressed. The paper is generally well written with nice graphics (apart from fig 7).

Although the analysis is decent, and it gives the reader an impression of the quality of the satellite HCHO column data, I do, however, feel that a somewhat opportunity has been missed since the aircraft data could have been used to really pick the retrievals apart. Maybe this is planned for the future, one can only hope!

Specific Comments:

Page 2, Line 25 & subsequent paragraph: Please note that Barkley et al (2013) compared OMI data to GABRIEL aircraft HCHO measurements and found OMI was about 40% too low.
We now cited Barkley et al [2013] in page 2, line 30–31.

Table 1 – I would include much more information about the details of the retrieval and AMF algorithms, e.g., fitting windows, absorbers fitted, use of radiance references, CTM vertical and horizontal resolutions, profile (monthly vs daily), resolution of scattering weight grids, treatment of clouds, terrain corrections, treatment of aerosols etc.
We now added Table S1 in the Supplementary Materials to summarize more retrieval details. We also referred to Table S1 in the text (page 3, line 15–16) and in the footnote of Table 1 (page 23, line 5).

Also discuss explcilty exactly how the reference sector correction is done in each case – yes they all follow the same approach - but there are subtle differences. This is a validation paper after all, so the reader needs to be well informed about the retrieval differences.
We now discussed the reference sector correction in the footnote of Table S1.

Page 3, Line 21: You have gridded the data yourself, so why quote the error on the monthly means from De Smedt's 2008 paper? Surely you can compute this error for each retrieval.
We cited De Smedt et al [2008] here to make the point that uncertainties in HCHO columns can be beaten down by temporal averaging. We have rewritten this part, please see page 3, line 28–29.

Equation (1) - please state if sigma_0 is a vertical (I assume it is) or a slant column background.
Sigma_0 (the reviewer meant to be omega_0) is a CTM-based vertical column. We have clarified this in the text (page 4, line 8).

Reference sector correction – since the retrieval groups are authors on the paper, I would urge the satellite data producers to provided their uncorrected slant columns (i.e. the raw retrieved slant

columns) so that a consistent a reference sector adjustment could be applied to all. This really would add substantial value to the paper. But I suspect that it will not be attempted, which is a shame.

Among all the 6 products, only BIRA retrievals provided the uncorrected slant columns. SAO and OMPS-PCA retrievals provided uncorrected vertical columns, which can be converted to uncorrected slant columns via AMF. We agree with the reviewer that retrieval groups should provide uncorrected columns in their products.

Page 4, line 27: "diurnal variability in the HCHO column is expected to be small..." Please quantify small – is this statement based on measurements or model studies?

We have rewritten this sentence, please see page 5, line 6–7.

Page 5: The normalisation process concerns me. I can sort of see why it was done but there are some opened questions.

We have rewritten this part, please see page 5, line 15–32.

For example, what are the spatial gradients in mixing layer depths, how does this impact the HCHO column.

We have normalized the impact of mixing layer depths on HCHO columns, please see page 5, line 16–18.

It is stated that day to day variability is "fitted well" by exp[0.11T] – please quantify well? What is the uncertainty in the comparison procedure due to fitting this function? How does the satellite comparisons change say if an arbitrary function of exp(cT) is used instead, where 'c' is a different constant?

We have added more details in the exponential fitting, see page 5, line 26.

Isoprene emissions, and thus HCHO columns, are very sensitive to temperature. What do you mean by 'local surface air temperature', where is this from (is it the GEOS-FP data) & how accurate is it?

"Local surface air temperature" refers to the "local surface air temperature at the time of the flight", please see page 5, line 29.

The conversion to a 'mean HCHO columns', using the mean vertical profile, again what is the error in this procedure and how does it impact the subsequent comparisons? What if you use a different assumed profile shape?

We assume here that the mean observed HCHO vertical profile (Figure 1, right panel) is the true profile.

Lastly, please discuss why it's really needed at all? Surely, you could compute HCHO columns for each flight, corresponding to a specific day and underlying surface footprint, and compare this to the satellite observations that lie in the same footprint (in which case you don't need to adjust for day-to-day variability)? I am, perhaps, being quite harsh but this step in the study is very important and yet it is given only a paragraph. Given the importance of the paper, I urge the authors to thoroughly discuss this key part of the analysis.

Single-scene HCHO retrieval has high uncertainty (~100%). So the direct comparison between coincide satellite pixel and aircraft point is not practical. The point here is to obtain the campaign-averaged satellite HCHO columns to beat down the retrieval noise. We have rewritten the text to clarify this, please see page 5, line 18–20.

Section 4: Why don't the retrievals apply the same GEOS-Chem model background correction, and probably better still, compute the AMFs using the GEOS-Chem model data? This would help isolate the columns differences due to the spectral fitting approaches. Yes, this would be some work but it would really add another dimension to the paper. You're already half way there, using the mean CAMS profile anyway.

This paper focuses on difference in HCHO columns among various products and potential drivers of the difference. So we just use the data provided by the retrieval groups without adding new information. Background corrections are almost the same (Table 3, column "$\Omega_o$"), this won't cause too much difference.

Page 8, lines 1-2: the AMFs are also sensitive to the respective cloud parameters for each retrieval (which instrument specific and subject to their own uncertainties), hence it is not just the scattering weights and angles.

We have rewritten this sentence, please see page 8, line 15–17.

Page 8, line 5 onwards: Again using the same surface reflectance data set in the AMF for all retrievals, would give some idea of its relative importance in this validation exercise. Ok, each retrieval 'should' use a reflectance data set consistent with the cloud algorithm but it would be an interesting experiment to carry out.

The reviewer made a good point. But here we focus on differences in HCHO columns among various products and potential drivers of the difference. We have cited sensitivity of AMF to surface albedo from De Smedt et al. [2008], please see page 8, line 22–23.

Page 8, line 15: Am I missing something here, but you've corrected the GEOS-Chem model using the CAMS data, which is adjusted for day-to-day variability, and yet this shows 'good agreement' with temporal dependence of isoprene emissions. Isn't this a somewhat circular comparison/argument? Please elaborate, and if not, then make sure you adjust the text to avoid leaving the reader with this impression.

The reviewer seems mistaken. We applied a +10% correction to the mean GEOS-Chem column averaged over the Southeast US during the entire campaign. We now clarified what we meant, please see page 8, line 27–34.

Page 8, line 30: I would say spatial correlations are 'moderate to relatively high', rather than give the feeling that the retrievals are better than they really are.

Accepted, please see page 9, line 17.

Page 9, final paragraph: yes you are right in what you say! But couldn't the authors address some of these issues in the framework of this paper?

We do give our recommendations, there is no more can be done in the framework of this paper. We have rewritten this paragraph to clarify what we meant, please see page 9, line 28–31.

Figure 3: are the temperature and HCHO columns, both sampled according to OMI observations (i.e. under the same cloud conditions?)

Temperature was the midday (1200–1300, local time) surface air temperature. GEOS-Chem HCHO columns were sampled according to OMI observations. We now clarified this, please see page 18, line 7–8.

Figure 5, the GEOS-Chem data is sampled along the 'OMI schedule' – according to which OMI retrieval? I suspect that each retrieval will have different quality flags for their measurements, so a 'good' measurement in one might be flagged as 'bad' in another. Are the actual same OMI pixels gridded between the retrievals? This also holds for the OMPS data as well. Maybe discuss this in the main text to clarify.

We now clarified this, please see page 20, line 4–5.

Figure 6: why not show the AMFs alongside????

We have added a panel in Figure 6 to show AMFs (w(σ)S(σ)) as functions of pressures.

I'm sorry but Figure 7 looks like coloured spaghetti. Maybe make it multi-panelled, with GEOS-Chem plus OMI data, them GEOS-Chem plus the OMPS data etc.

We are now only showing OMI-SAO and OMI-BIRA in Figure 7. We mentioned other retrievals in the text (page 8, line 30–33).

Table 1 please explicitly state how the detection limits are calculated for each retrieval.

We now stated how the detection limits are determined for each retrieval in the text (page3, line 23–28).

**Anonymous Referee #2**

General Comments This paper presents intercomparison results for HCHO vertical columns measurements from 4 satellite instruments and 6 different retrievals and comparison to an aircraft campaign over the Southeast US. The paper is easy to read and very interesting for the community in order to have a better view of the quality of the existing satellites HCHO retrievals. It is within the scope of ACP and should be published after taking into account the comments and improvements listed below.

Specific Comments

page 3, line 3: it's the only point in the paper where the in-situ data (reference and technique) are introduced, and as the whole paper is based on these measurements, it would be good to say a little more about them (maybe at the beginning of Section 3). What is their accuracy, sensitivity, ... see comment relative to page 4.
We have added two sentences to describe the two instruments, please see page 3, line 6–8.

page 3, line 10: to my view, Table 1 should contain more information on the different retrievals. It should highlight the main differences between the various retrievals made on the same satellite sensor. Other differences than only the CTM are important: the albedo, surface, cloud correction, etcc (that are only mentioned in page 8) that impact AMF calculation, but also differences in DOAS settings for the slant column retrieval.
We now added Table S1 in the Supplementary Materials to summarize more retrieval details. We also referred to Table S1 in the text (page 3, line 15–16) and in the footnote of Table 1 (page 23, line 5).

page 4, line 16: and applied Equation (1) "to" compute ... page 4, line 17: AMF based on the its reported: remove "the" or "its".
Accepted, please see page 4 line 25–26.

page 4, section 3: I would add a small paragraph on the 2 in situ techniques that are mentioned in the introduction, and that are considered the "truth" but that have 10% difference, as stated in line 31. Any hint on why?
We have added two sentences to describe the two instruments, please see page 3, line 6–8. The 10% difference is due to the fact that the two instruments are independently calibrated. This difference is generally within the mutual stated accuracy for both instruments. We now explained this difference in the text, please see page 5, line 9–12.

The organization of the figure sequence in this section is a little bit confusing for me (the different part of each figures are not discussed/mentioned together, but at different moments of section 3): first figure 1 (but only right and upper left panels), then Figure 2 (left panel), then bottom left panel of figure 1 (p.5, line 18), then figure 3 (p.5, line 20), then missing reference to figure 4 (p.5, line 29) and then only in page 6, the right panel of figure 2 is presented/mentioned. Maybe the bottom left panel of figure 1 could go with figure 3 and the right panel of figure 2 could go after figure 4?

We have considered rearrangement of those panels, but we would like to stick to the same layout.

page 4, line 24: it is stated that the typical extension of the mixed layer is to 1.5-2 Km in the afternoon, but in figure 1, the caption mention 1 to 3km. It would be good to add the height scale in km on the profile picture (and not only pressure) in order to easily make the link with the figure.
We have changed "1.5–2 km" to "1–3 km", please see page 5, line 3. We have also added a height scale (in km) in Figure 1.

page 4, line 26: most flight hours were in the afternoon. How much is "most"?
We have rewritten this sentence, please see page 5, line 5–6.

page 4, line 27: the diurnal variability is expected to be small. How much is "small"?
We have rewritten this sentence, please see page 5, line 6–7.

page 5, line 5: considering the 10% difference between the 2 in-situ technique, why is CAMS considered as the reference in the rest of the paper (GEOS-CHEM is increased by 10% to reach CAMS values) and not ISAF or the average of both?
We chose to use CAMS as reference because it is less biased against GEOS-Chem (Figure 1 and 2), but we also commented on the impact of using ISAF in the text. We have rewritten several sentences to clarify this, please see page 5, line 12–14.

page 5, line 29: add "as can be seen in figure 4, " before "Initial simulations ..."
Accepted, please see page 6, line 10.

page 5, line 31: add "previous"before "comparisons of GEOS-FP ..."
Accepted, please see page 6, line 12.

page 6, line 1: add "red line in" before "figure 4".
Accepted, please see page 6, line 15.

page 6, line 11: how do you "remove this dependence on altitude"? again, it would be helpful for an easy reading of the paper, to mention the limits of the mixed layer and of the free troposphere in pressure values.
We have rewritten this sentence, please see page 6, line 27. We have also added a height scale in Figure 1, and mentioned the limits of the mixed layer (page 5, line 3) and the free troposphere (page 6, line 30) in pressure in the text.

page 6, line 18: why only mention the spatial correlation with CAMS data and not with ISAF?
We have added the spatial correlation coefficient with ISAF data, see page 7, line 1–2.

page 6, line 26 and lower and Table 2: the spatial (and temporal) correlation coefficient calculation should be mentioned in the text here, and not only in the caption of Table 2
Accepted. We have rewritten this sentence, see page 7, line 9–10.

page 6, line 29: GOME-2A BIRA is noisier wrt GOME-2B, this is clear. How much is this due do degradation and how much to the reduced swath mode?

We found that noise of GOME2A columns is significantly the same before and after the swath mode reduction, so we attribute the noisier pattern seen by GOME2A to its instrument degradation rather than its reduced swath mode. We have rewritten the text to clarify this, please see page 7, line 12–15.

page 7, line 2: "GEOS-Chem columns are sampled on the same schedule and scenes as the individual retrievals" but as GOME2A and GOME2B does not have the same swath, the pixels geometry and cloud impact are not the same, so why the value of GEOS-Chem columns in table 3 is the same (1.59) for both instruments?
We have fixed this in Table 3.

page 7, line 8 to 14: This would be a good place to discuss the different retrieval choices and their impacts on the different retrieval steps (corrected slant column, AMF and background correction).
Actually, we did start to discuss impacts of retrieval choices on retrievals from this paragraph. Please see page 7 line 23–28 for corrected SCD, page 8 line 1–23 for AMF, page 8 line 24–26 for background correction.

page 7, line 22: after discussing the OMI-BIRA shape factors (ie the IMAGES input) wrt to CAMS profile, you could discuss the other model shapes too? and how are GEOS- CHEM and IFAS profile? is this difference between model expected/known? what is the possible difference explanation?
We have rewritten this, please see page 8, line 4–9. The discrepancies between GEOS-Chem and CAMS (and ISAF) are likely due to the insufficient deep convection in the model, we have mentioned this in the text (page 6, line 30).

page 7, line 29: the "much smaller" affirmation in the sentence "The differences with the original AMFs are less than 6.0%, a much smaller correction than for OMI-BIRA" would be more clear with "are less than 0.06, while for OMI-BIRA the change in AMF is around the double (-0.14)".
We have rewritten this sentence, please see page 8, line 10–11.

page 8, line 2: not only "driven by scattering weights and viewing angles". The clouds, albedo, aerosols also impacts the AMFs.
Accepted. We have rewritten this, please see page 8, line 15–17.

page 8, line 5: the difference in surface reflectivity between OMI-SAO and OMI-BIRA is not so small (~23 to 29%). this certinaly impact (and not only "would contribute in part) "the difference in scattering weights ar lower altitudes". Please refer to sensitivity tests and AMF error estimations from literature.
Accepted. We have rewritten this part, please see page 8, line 21–24.

page 8, line 8: the sentence "The scattering weights of OMI-BIRA are lower than those of GOME2B-BIRA (Figure 6) and GOME2A-BIRA (about the same as GOME2B-BIRA, not shown), even though all BIRA retrievals use the same surface reflectivity" is mis- leading. The retrieval is the same (same surface reflectivity), but the geometry of the 2 instruments is different, as well as the time of the day and clouds", so we expect different scattering weights.
We have removed this sentence from the text, please see page 8, line 21.

page 8, line 14: a discussion of table 3 wrt to differences in the slant columns values between the different retrieval is missing (discussion on VCD, AMF and background correction has been done before).
Actually, we did discuss the corrected slant column along with the AMF in page 7, line 22–28.

page 8, line 18: what is the difference between the temporal correlation reported in figure 7 (and here in the text) and temporal correlation values of table 2 ?.
Figure 7 shows the temporal correlation between satellite HCHO column and temperature, please see page 8, line 31–33. Table 2 summarizes the temporal correlation coefficients between HCHO columns for different pairs of satellite retrievals, computed from daily averages of each retrieval over the Southeast US domain. Please see page 7, line 8–9 for description of Table 2.

page 9, line 9: I find the sentence "Aside from OMI-BIRA, the shape factors used in the retrievals are not a significant source of error" a little bit odd. If I understand well table 3, the difference in VCD bias when considering the original profile and when using the CAMS profile is 8% (from -20% to -12% bias) for OMI-BIRA, which is indeed the highest change, but the other products ranges between 7% (for GOME2B-BIRA), to 6% (OMI-SAO), 5% (OMPS-SAO), 3% (GOME2A-BIRA) and finally 1% (OMPS-PCA), which are not so different than 8%. Please reformulate this sentence.
We now clarified this, please see page 9, line 26–27.

**Anonymous Referee #3**

This manuscript presents an intercomparison exercise of tropospheric HCHO retrieved from satellites measurements using several products obtained from independent retrieval approaches. In order to validate the products, the authors use aircraft observations during the short term field deployment of SEAC4RS in the Southeast of the US. This is an important validation effort. In general, the paper is well written and within the scope of ACP. That said, I have important comments/suggestion that I recommend revisions before final publication.

Major comments:

- In the abstract and conclusion sections it is mentioned that the HCHO columns are biased low by 20-50%, which is a significant number. Then, it is concluded that HCHO from satellite provides a reliable proxy for isoprene emissions. How do the authors conclude that HCHO is reliable given the significant bias (plus any uncertainty associated to the assumptions of HCHO yields from isoprene conversion)?. The only section with results shown is section 4, I recommend to insert a new section(s) where specif details are given regarding errors and how they affect the isoprene emission.

The 20–50% bias is systematic among retrievals and we gave those correction factors based on in situ observations. We have added one paragraph to discuss effect on isoprene emission estimation, please see page 9, line 1–6.

- The emissions of natural biogenic volatile organic compound have a temperature dependence, hence a seasonal variation that it is not studied in the present study. How- ever, the statements along the manuscript are quite general regarding spatial and temporal distribution of HCHO. The focus of the study is only limited in the southeast of the US during less than two months. I suggest to state clearly that the results/conclusions shown are for this specific time/area.

We have specified this in multiple places in the text.

- The aerosol optical depth (AOD) observed in the southeast of the US is significantly higher than other parts in north America. In addition, the temporal distribution of AOD over the southeast of the US has a maximum peak in the summer months (similar months as this study). However, it is not well documented in section 2 how aerosols are treated in the satellite retrievals. Also, current investigation in the southeast of the US associate aerosol aloft (e.g., Goldstein et al., 2008). I encourage the authors to explain in better detail the effect of aerosols in the retrieval and final uncertainty. How do aerosol aloft impact the retrieval of HCHO columns?

"Impact of aerosols is not explicitly addressed in HCHO retrievals because it is considered to be implicitly included in the cloud correction scheme to the scattering weights." We have added this sentence in page 4, line 20–21.

- The in-situ sensors are taken as the ground truth in this study, however there is quite a bit of manipulation in the conversion of mixing ratios to columns. It is not clear to me why the HCHO need to be normalized if the columns are compared with satellite retrievals. Please give a thorough description of why this is needed and why the "Day-to-day variability in HCHO columns can be fitted well to exp[0.11T]".

We have rewritten the normalization part, please see page 5, line 15–32.

Is there any explanation of why CAMS and ISAF have differences of about 10% since they measure the same air mass?
The 10% difference is due to the fact that the two instruments are independently calibrated. This difference is generally within the mutual stated accuracy for both instruments. We now explained this difference in the text, please see page 5, line 9–12.

- Have the authors looked at the trace gas inhomogeneities captured by the in-situ observations and compare with results of satellite on a pixel-pixel resolution and ?. In other words, do the correlation improve if the air mass sampled is the same?
We didn't focus on individual pixel validation in this paper, because individual satellite has high uncertainty. To reduce the uncertainty, this validation work was done on monthly (Aug. 05 to Sep. 25) and regional (Southeast US) average basis.

- In the abstract it is mentioned that "The GEOS-Chem chemical transport model pro- vides a common intercomparison platform". However, it is not clear how GEOS-Chem is used. Consider expanding the sentence explaining how this is achieved.
We have rewritten this sentence, please see page 1, line 27–28.

- Page 3, line 23: "Here and elsewhere, we use only satellite pixels with solar zenith angle less than 60, cloud fraction less than 0.3, and row anomalies (for OMI) screened". In the same paragraph it is mentioned the detection limit of the satellites but it is not clear if you use only data above the detection limit for the analysis in this work.
We didn't filter satellite pixels by detection limits. Instead, we discarded pixels with too high ($10 \times 10^{16}$ molecules cm$^{-2}$) or too low (-0.5$\times 10^{16}$ molecules cm$^{-2}$) column densities. We have clarified this in this text, please see page 3, line 31–32.

- Page 4, line 16: There are several abbreviations that need to be defined, e.g., SAO, OMPS-PCA, etc. A table/appendix with abbreviations would be useful.
We now summarized those abbreviations in the Appendix. Please see page 10.

- Page 5, line 5-17: The conversion of mixing ratios to columns is achieved by assuming that HCHO is co-located with aerosols (identified with the mixing height from DIAL- HSRL), how valid is this assumption?. I suggest to explain also why an exponential decay with a scale height of 1.9 is used and how do the background column is found.
We have rewritten this paragraph, please see page 5, line 15–32.

[revised manuscript text omitted]

Formatted Table

| Page 5: [1] Deleted | Zhu, Lei | 7/8/16 11:13:00 AM |

Part of the horizontal

| Page 21: [2] Deleted | Zhu, Lei | 7/8/16 11:13:00 AM |

[Figure]

Figure 6. Mean

| Page 21: [3] Deleted | Zhu, Lei | 7/8/16 11:13:00 AM |

resulting air mass factor (AMF) computed

| Page 21: [3] Deleted | Zhu, Lei | 7/8/16 11:13:00 AM |

resulting air mass factor (AMF) computed

| Page 21: [3] Deleted | Zhu, Lei | 7/8/16 11:13:00 AM |

resulting air mass factor (AMF) computed

**Page 21: [3] Deleted**          **Zhu, Lei**          **7/8/16 11:13:00 AM**

resulting air mass factor (AMF) computed

**Page 23: [4] Deleted**          **Zhu, Lei**          **7/8/16 11:13:00 AM**

[c] Single-scene spectral fitting uncertainties (except for OMPS-PCA). Fitting windows for the different retrievals are in the 327.7–356.5 nm range. Detection limits of GOME2A and B were in 2013. Detection limit of OMPS-PCA is determined as the 4-times of 1-sigma noise over the Pacific Ocean [Li et al., 2015].

[d]

---

## Author Response (AR2)

We thank the reviewer for the thorough and thoughtful comments. Our responses to the reviewer are in blue.

Second Review of Zhu et al.

Specific issues:

Page 3 Lines 9-10. You imply that direct satellite validation profiles are not useful, as the noise in the satellite data is high. This is not really true is it? One can still average validation profiles coincident with satellite observations, to mitigate for noise in the retrievals.
We have rewritten this sentence to avoid ambiguity. Please see page 3, line 9–11.

Page 3, lines 28-29: The gridded HCHO errors on the de Smedt et al (2008) correspond to GOME & SCIAMACHY averages, which have sample less frequently than the sensors in this study. Please give the appropriate errors for OMI, GOME-2 (A/B) and OMPS – as asked for originally.
We have listed error associated with each retrieval. Please see page 3, line 28.

Page 3, lines 31-32. Why was data only in the range -0.5 to 10 x1016 molecules/cm2 used?
We have clarified this. Please see page 4, line 1–2.

Page 5, lines 6-7. It would be more accurate to say "Diurnal variability in the HCHO columns over this regions is expected from models to be less than 10%, assuming a correctly simulated diurnal photochemical cycle…" (or similar).
Accepted. We have rewritten this sentence, please see page 5, line 6–7.

Page 5, lines 18-20. Okay, you construct mean HCHO columns to reduce "noise" (here I assume you mean in the retrieval). However, I still think computing the 'aircraft' HCHO columns for each flight and their spatial extent, and comparing that to the satellite data that lie within the aircraft footprint, should be a first step. OMI has good coverage, so you should still get a sufficient number of observations to average. Even if the comparison is 'bad' as you indicate, it would still be useful to show it maybe in the supplementary materials (& if the approach is poor then it of course adds credibility to your chosen method).
This would be too noisy. Averaging data prior to comparison is standard practice in satellite validation studies, in particular for HCHO which is so noisy.

Page 5, lines 20+. The uncertainty in the conversion using the mean HCHO profile has not been quantified. Assuming the mean observed HCHO profile, is a good approach, but it still needs some error associated with it (yes, the error maybe small but it needs to be in the paper). This was asked for in the previous review & should have been included in the revised manuscript, but for some reason the authors have not done it.
Error in the mean HCHO profile is negligible: CAMS has a precision of 40 ppt, then the error in the mean mixing ratios at each pressure bin is tiny (1–5 ppt) by averaging. "We estimate the error in this mean HCHO columns is ~15%, which is mainly from the mixing depths, assumed background, scale height and temperature dependence." We have added this sentence in page 5, line 34–page 6, line1.

Section 4. Again I still think computing consistent AMFs using GEOS-Chem would be a useful exercise and would help to identify and explain the retrieval differences, but it shouldn't limit publication.

Actually, we did compute AMFs using GEOS-Chem profiles, but we realized that working with a single CAMS profile has the advantage of being more transparent. Therefore, we just apply the same observed CAMS mean profile (assumed to be true) to all the six retrievals to identify other possible factors of the retrieval differences. We have included a figure in the Supplemental Materials to show GEOS-Chem based AMFs. We have also added a sentence in the text to that effect, please see page 8, line 14–15.

One final point: There are several places were you have written "HCHO columns serve as a useful proxy isoprene emissions" (or similar). It should be something like "our analysis/results show that HCHO columns may serve as a useful proxy isoprene emissions over this region" – although we already knew this based on the works of Palmer et al and Millet et al.!

We understand the reviewer's point and have modified the text in the several places. We don't want to say "over this region" because there is no reason to presume that our results would not have generality.

[revised manuscript text omitted]

[a] SAO retrievals applied the "pixel-by-pixel correction" method, which accounts for the difference between retrieved and GEOS-Chem modelled slant columns over the remote Pacific Ocean. For BIRA retrievals, the reference sector correction refers to the latitudinal dependency (modelled by a polynomial) of the HCHO slant columns in the reference sector. For OMPS PCA retrievals, the latitude-dependent GMI monthly climatology HCHO column amount over the remote Pacific is added to all pixels, without pixel-specific correction.

[Figure]

[Figure]

**Figure S1**. GEOS-Chem based air mass factors (AMF) over the Southeast US averaged over the SEAC[4]RS period (5 August–25 September 2013). Each panel shows the AMF computed using GEOS-Chem HCHO shape factors with scattering weights from the six retrievals (Table 1). For each retrieval, GEOS-Chem shape factors are sampled under its schedule, and filtered by its quality flags and cloud conditions.